# Nanoemulsion Increases the Antifungal Activity of Amphotericin B against Four *Candida auris* Clades: In Vitro and In Vivo Assays

**DOI:** 10.3390/microorganisms11071626

**Published:** 2023-06-21

**Authors:** Gabriel Davi Marena, Alba Ruiz-Gaitán, Victor Garcia-Bustos, María Ángeles Tormo-Mas, Jose Manuel Pérez-Royo, Alejandro López, Patricia Bernarbe, María Dolores Pérez Ruiz, Lara Zaragoza Macian, Carmen Vicente Saez, Antonia Avalos Mansilla, Eulogio Valentín Gómez, Gabriela Corrêa Carvalho, Tais Maria Bauab, Marlus Chorilli, Javier Pemán

**Affiliations:** 1Severe Infection Research Group, Health Research Institute La Fe, 46026 Valencia, Spain; 2Department of Drugs and Medicines, School of Pharmaceutical Sciences, São Paulo State University (UNESP), Araraquara 14800-903, Brazil; 3Department of Biological Sciences, School of Pharmaceutical Sciences, São Paulo State University (UNESP), Araraquara 14800-903, Brazil; 4Department of Medical Microbiology, University and Polytechnic La Fe Hospital, 46026 Valencia, Spain; 5Department of Pathology, Faculty of Medicine and Dentistry, University of Valencia, 46010 Valencia, Spain; 6Department of Pathological Anatomy, La Fe Hospital, 46026 Valencia, Spain; 7Department of Microbiology and Ecology, University of Valencia, 46010 Valencia, Spain

**Keywords:** *Candida auris*, *Galleria mellonella*, nanoemulsion, amphotericin B, infectious disease

## Abstract

*Candida auris* is an emerging yeast of worldwide interest due to its antifungal resistance and mortality rates. The aim of this study was to analyse the in vitro and in vivo antifungal activity of a nanoemulsion loaded with amphotericin B (NEA) against planktonic cells and biofilm of *C. auris* clinical isolates belonging to four different clades. In vivo assays were performed using the *Galleria mellonella* model to analyse antifungal activity and histopathological changes. The in vitro results showed that NEA exhibited better antifungal activity than free amphotericin B (AmB) in both planktonic and sessile cells, with >31% inhibition of mature biofilm. In the in vivo assays, NEA demonstrated superior antifungal activity in both haemolymph and tissue. NEA reduced the fungal load in the haemolymph more rapidly and with more activity in the first 24 h after infection. The histological analysis of infected larvae revealed clusters of yeast, immune cells, melanisation, and granulomas. In conclusion, NEA significantly improved the in vitro and in vivo antifungal activity of AmB and could be considered a promising therapy for *C. auris* infections.

## 1. Introduction

*C. auris* has some characteristics that distinguish it from other *Candida* species: (i) it is difficult to identify as it is often misidentified by conventional diagnostic techniques; (ii) different antifungal resistance profiles have been observed and it has become multi-drug resistant; (iii) its prevalence is unknown both in the general population and in environmental niches; and (iv) the mechanisms of dissemination are unclear, making control difficult [1,2].

Several studies have described the involvement of *C. auris* in candidemia and other invasive infections with high mortality and morbidity worldwide [3,4]. This emergence of cases can be attributed to their survival in the environment, rapid colonisation of the patient’s skin, and high transmissibility in the hospital environment. This has led to severe and prolonged nosocomial outbreaks. Regarding the environment, its ability to persist on dry and wet surfaces (floors, sinks, beds) for an average of seven days is alarming. Studies analysing the ability of *C. auris* to colonise patients have reported that this species can be shed from the skin at a rate of approximately 10^6^ cells/h, and it is estimated that approximately 4 h is the minimum contact time for yeast acquisition [5,6,7,8].

Because of its multi-resistance, the therapeutic options for systemic *C. auris* infections are limited. Therefore, there is a need to find therapeutic alternatives for the management of infections caused by this pathogen [9].

Among the various nanotechnological strategies in the field of drug delivery systems, nanoemulsions (NEs) are transparent systems in which the oil phase is dispersed in the aqueous phase (or vice versa) with the aid of a surfactant that promotes the reduction of interfacial tension or the surface that exists between these two liquids [10,11]. NEs have small droplet sizes, good stability, a long shelf life, low energy requirements for production, and an easy formation process, thus facilitating the incorporation of poorly soluble drugs. NEs contribute to the safe and effective absorption and penetration of drugs at the site of action [12]. Notably, in an NE, substances can be incorporated into the oil phase or water phase depending on the affinity of the substance under study. Furthermore, this system allows the co-encapsulation of drugs with different solubility profiles [13].

Thus, the encapsulation of amphotericin B (AmB) in NE could help increase its therapeutic effect. NE can provide better bioavailability and direct action against pathogens at lower doses and with reduced toxic effects due to its high selectivity. Therefore, this study aimed to determine the potential antifungal activities of free AmB and AmB encapsulated in nanoemulsions against *C. auris* clinical isolates. Furthermore, the use of NE is expected to enhance the antifungal activity of AmB against different clades of *C. auris* in both planktonic and sessile cells and also in an in vivo *G. mellonella* infection model.

## 2. Materials and Methods

### 2.1. Development and Characterisation of the Nanoemulsion

The NE was developed according to the protocol previously described by Marena et al. [14,15].

### 2.2. Fungal Strains

The antifungal activity of AmB and NEA was evaluated against four clinical isolates of *C. auris* belonging to four different clades: *C. auris* VPCI479/P13 (India, CLADE I—InP13), *C. auris* CBS10913 (Japan, CLADE II—JAP 1), *C. auris* CBS 15603 (Spain, CLADE III—SP96), and *C. auris* VEN C6 (Venezuelan, CLADE IV).

### 2.3. Determination of the Minimum Inhibitory Concentration (MIC)

MIC determinations were performed according to the CLSI M27-A3 document [16]. Briefly, AmB was dissolved in dimethyl sulfoxide (DMSO) 5% + Roswell Park Memorial Institute 1640 (RPMI) medium (Sigma Aldrich, North Rhine-Westphalia, Germany), and NEA was diluted in RPMI. Serial dilutions were performed in RPMI supplemented with dextrose 2% in the concentration range of 0.0019–5 µg/mL.

The controls used in each assay were as follows: NE without AmB + inoculum; solvent (RPMI + 5% DMSO) + inoculum to demonstrate no antifungal activity; growth control (inoculum + RPMI) and sterility control (AmB and NEA + RPMI). The MIC_50_ and MIC_90_ values (MICs at which 50% and 90% of the isolates were inhibited, respectively) were calculated. All plates were measured spectrophotometrically, and the data were analysed using GraphPad Prism 8.0. The experiment was performed at least three times on three different days. To calculate the MIC_50_, the absorbance data from the three tests were analysed using GraphPad Prism 8.0 software. The standard deviation corresponds to the analysis of the means of the three tests.

### 2.4. Determination of Antibiofilm Activity

The biofilm formation method described by Pitangui et al. [17] was used, with some modifications. Briefly, yeasts cells were grown in 96-well microplates in yeast extract peptone dextrose (YEPD) broth medium (Scharlab SL, Barcelona, Spain). For pre-adhesion, the cells were inoculated for 2 h without rotation. Two types of biofilms were evaluated: biofilm in formation (24 h) and mature biofilm (48 h). The controls used in each assay were growth control (inoculum + YEPD), NE without AmB + inoculum + YEPD (to determine that the NE had no antibiofilm effect), solvent control of 5% DMSO + YEPD + inoculum (to verify that the solvent had no antibiofilm effect), sterility control (YEPD without inoculum), and NEA and AmB without inoculum.

#### 2.4.1. Biofilm in Formation

A 96-well microplate was inoculated with 1 × 10^6^ cells/mL for 2 h. After incubation, the supernatant containing the non-adherent cells was removed with 200 µL of PBS and the adhered cells were treated with 100 µL of AmB or NEA in YEPD at concentrations between 0.07 and 20 µg/mL and incubated at 37 °C for 24 h. After incubation, the microplates were washed with 200 µL of PBS. To determine the metabolic activity, 100 µL of 2,3-bis(2-methoxy-4-nitro-5-sulfophenyl)-5-[carbonyl(phenylamino)]-2H-tetrazolium-hydroxide (XTT^®^ at 0.005 g/10 mL—Thermo Fisher Scientific, Waltham, MA, USA) was added to each well. The plates were incubated at 37 °C for 2 h in the dark and spectrophotometric readings were performed at 492 nm. The experiment was conducted at least thrice on different days.

#### 2.4.2. Mature Biofilm (48 h)

After the pre-adhesion time, the non-adherent cells were removed using 200 µL of PBS. Next, 100 µL of YEPD was added to each well and incubated at 37 °C for 48 h. The biofilm formed was washed with 200 µL of PBS, and 100 µL of AmB or NEA at 0.78 to 50 µg/mL was added. Then, the plates were incubated at 37 °C for 24 h. After incubation, the biofilm was washed again with 200 µL PBS, and the metabolic activity was measured using the XTT method, as described in Section 2.4.1. The experiment was performed at least three times and on different days.

### 2.5. Antifungal Assay Using an In Vivo Model of Galleria mellonella

The day before infection, healthy cream-coloured *G. mellonella* larvae of 250–350 mg (Dnatecosistemas, Madrid, Spain) were placed in a plastic container with ventilation holes and incubated at 37 °C for adaptation. On the day of infection, the larvae were kept in a Petri dish with ice for 2–3 min for immobilisation (groups of 20 larvae). Prior to inoculation, the larva prolegs were disinfected with a 70% alcohol-impregnated swab. Then, using a Hamilton Microliter™ syringe, 10 µL of *C. auris* inoculum (10^4^ cells/larva) resuspended in PBS + AmP (Ampicillin, Sigma Aldrich, Steinheim, North Rhine-Westphalia, Germany) at a concentration of 20 µg/mL was administered into the penultimate proleg of each larva [18]. The larvae were incubated at 37 °C for 2 h. AmB was solubilised in PBS + AmP + DMSO (0.05%), and NEA was solubilised in PBS + AmP. After 2 h, 10 µL of AmB and NEA (1.68 mg/Kg—3× MIC) was injected into the penultimate proleg. The treatment was carried out every day for 5 days. Controls for the *G. mellonella* in vivo assay were as follows: (i) the group without treatment and without infection (negative group), (ii) the group infected and treated with PBS + AmP (diluent control), and (iii) the group infected and treated with NE without the drug (to confirm that NE has no antifungal activity). All the larvae were incubated at 37 °C for 24 h. The syringe was immersed in 70% ethanol for 2–3 min, rinsed five times with 70% ethanol, and then rinsed five times with sterile 1× PBS between the strain procedures.

After 24 h of incubation, three larvae from each group were selected for total haemolymph extraction by decapitation and tissue collection. The haemolymph was collected in Eppendorf tubes, diluted (1:2, 1:10, 1:100) in PBS + AmP (20 µg/mL), and cultured on Sabouraud agar + chloramphenicol (PanReac AppliChem, Barcelona, Spain). The tissue was transferred to tubes containing 2 mL PBS + AmP and triturated in Ultra Turrax^®^ T25 (Janke & Kunkel IKA^®^—Str. 10 79219 Staufen / Germany and cultured on Sabouraud agar + chloramphenicol after dilution in PBS + AmP (1:10, 1:100, and 1:1000, respectively). All plates were incubated at 37 °C for 48 h, and colony counts were expressed as CFU/mL of haemolymph or CFU/mL of homogenised tissue. This procedure was repeated for five days. Three larvae from each group were sacrificed after 2 h of incubation and their haemolymph and tissues were cultured to determine the number of CFUs after infection. This result is expressed as time zero.

### 2.6. Histopathology

The larvae were infected and treated as described in Section 2.5. This assay consisted of the following groups: (i) non-infected larvae without treatment (negative control), (ii) infected larvae treated with PBS + AmP (infection control group), (iii) infected larvae treated with NE without the drug (NE infection control group), (iv) larvae infected and treated with AmB (1.68 mg/Kg), and (v) infected larvae treated with NEA (1.68 mg/Kg). The treatment was performed daily for five days. The fixation and staining methodologies were as described by Garcia-Bustos et al., with some modifications [19]. The larvae were collected 120 h after the first infection, anesthetised with 5% ethanol, and transferred to tubes containing 10 mL of 4% formalin for preservation. The larvae were fixed for at least 20 days. After fixation, the larvae were sectioned sagittally, and paraffin-embedded (FFPE) tissue blocks were stained with haematoxylin–eosin (HE) for the morphological analysis of organelles, granulomas, cell infiltration, and immune response. Tissues were also stained with periodic acid–Schiff (PAS) to assess tissue morphology, granulomas, yeast morphology, and yeast infiltration into tissues (biofilms). To quantify granulomas, three larvae from each group were collected 24, 48, 72, 96, and 120 h post-infection. The complete processes of fixation, sectioning, and staining were performed as previously described. Granulomas were quantified during histopathological analysis. Images were analysed under an optical microscope at 100× magnification and captured using a Xiaomi Mi 9T camera (Beijing, China).

### 2.7. Statistical Analysis

Statistical analyses were performed using GraphPad Prism 8.0. For the in vitro (biofilm) assays, a one-way analysis of variance (ANOVA) was used to compare the treatment with growth control. For the in vivo assays, a two-way ANOVA was used to determine intragroup differences and to compare between groups over five days of treatment. Tukey’s test correction was used for multiple comparisons between the groups. A *p*-value < 0.01 was indicated in each statistical analysis.

## 3. Results

### 3.1. Determination of the Minimum Inhibitory Concentration (MIC)

The AmB and NEA MICs are shown in Table 1. The MIC_90_ of NEA was considerably lower than that of AmB for VEN C6 (MIC_90_ 1.25 vs. 0.31 µg/mL), JAP 1 (MIC_90_ 0.31 vs. 0.038 µg/mL), and SP96 (MIC_90_ 0.62 vs. 0.038 µg/mL). Additionally, the NEA MIC_50_ values were lower. No fungal growth was observed in any of the control groups.

### 3.2. Determination of Antibiofilm Activity

#### 3.2.1. Biofilm in Formation

When evaluating the ability of NEA and AmB to inhibit biofilm formation, NEA showed a significant inhibition potential compared to AmB against InP13 and VEN C6 strains at concentrations of up to 0.07 µg/mL (*p* < 0.002 and <0.0001, respectively) (Figure 1A and Figure 1D, respectively). In contrast, NEA did not improve AmB activity against the JAP 1 and SP96 strains (Figure 1B,C).

#### 3.2.2. Mature Biofilm

Figure 2 shows the results of the NEA and AmB treatments against mature *C. auris* biofilms. NEA was more effective than AmB in inhibiting mature biofilms in all isolates. The analysis of the metabolic response by clade, NEA, and AmB at low concentrations (6.25 µg/mL; *p* < 0.0001) showed metabolic inhibitions of 9.6 and 38.6%, respectively, against the VEN C6 strain. The metabolic inhibitions by AmB and NEA at the same concentration were 34.5 and 58.6%, respectively, against isolate SP96.

For strains belonging to clades I and II, such as InP13, the metabolic inhibition obtained with AmB for the mature biofilm was 17.2%, whereas NEA showed a metabolic inhibition of 38.2% at the lowest concentration tested (6.25 µg/mL) (Figure 2). Similarly, for JAP 1, NEA was more active with a metabolic inhibition of 51.5% compared to 80.1% for AmB.

The present results suggest that NEA can preserve the antifungal activity of AmB, even at low concentrations. In contrast, the ability of free AmB to inhibit mature biofilm formation was significantly reduced at lower AmB concentrations. Notably, the SP96 strain (clade III) was the most susceptible to both treatments, showing significant metabolic inhibition at concentrations as low as 1.56 μg/mL (*p* < 0.0001) (Figure 2). Moreover, no significant differences were observed between the activities of AmB and NEA against VEN C6 at higher concentrations of 12.5, 25, and 50 μg/mL (Figure 2).

#### 3.2.3. In Vivo Antifungal Activity

The antifungal activity of NEA on the haemolymph and tissue in the *G. mellonella* model is shown in Figure 3 and Figure 4, respectively.

In general, NEA showed better antifungal activity in the haemolymph than the AmB-treated groups. When analysing the response by clade, the InP13-infected larvae group showed a greater reduction in fungal load with NEA treatment from day 2 (*p* < 0.05). There was no difference between the infection group (treated with PBS + AmP) and the AmB group (Figure 3A).

NEA was significantly more effective in reducing the fungal load in JAP 1 (clade II), and this difference was most evident on the third day of treatment (*p* < 0.005). Notably, NEA completely inhibited fungal load in the larval haemolymph (Figure 3) on day 5 of infection. In the group infected with SP96 and VEN C6 strains (clades III and IV, respectively), NEA reduced the fungal load to 10^2^ UFC/mL of haemolymph compared with AmB (10^3^ UFC/mL) by the last day of treatment (Figure 3).

In all larvae, a higher fungal load was observed in the tissues compared to the haemolymph. Differences in NEA and AmB activity were also observed. After five days of treatment, the infected group (InP13, SP96, and VEN C6 strains) and those treated with AmB had a fungal load of 10^7^, 10^6^, and 10^6^ CFU/mL tissue homogenate, respectively, whereas with NEA treatment, the fungal load was lower, with counts of 10^6^, 10^5^, and 10^5^ CFU/mL tissue homogenate for each strain (*p* < 0.05, *p* < 0.0001, and *p* < 0.0001, respectively) (Figure 4). In summary, NEA treatment demonstrated superior efficacy against all clades in *G. mellonella* tissue, particularly after three days of treatment (Figure 4). Although initial effectiveness was observed against the JAP1 strain on day 2 with AmB treatment (Figure 4), and against VEN C6 on days 3 and 4 (Figure 4), a fungal burden increase was observed during the later stages of treatment, indicating a potentially low sensitivity of the strains to the unbound drug.

#### 3.2.4. Histopathology

Figure 5, Figure 6, Figure 7 and Figure 8 show the histopathological responses to fungal infection after treatment with NEA and AmB. The images show the formation of clusters of yeast (L) surrounded by haemocytes (H), forming granuloma structures throughout the larval tissue. A large amount of melanin is produced during infectious and inflammatory processes, forming orange/brown regions (M). Granulomas are found throughout the body of the larva, being more abundant in the lower extremities (tail), upper extremities (head), and close to the trachea. Granulomas are found in different regions, surrounded by adipose tissue (at), haemolymph (H), or adhered to the walls of organelles or superficial cuticles (C).

In the larva group infected with InP13, SP96, and VEN C6 strains, the granulomas were larger, with a large amount of melanin, necrotic regions, and haemocytes (Figure 5, Figure 7 and Figure 8). In contrast, the larvae infected with the JAP1 strain showed granulomas of smaller (Figure 6E,F) or fragmented size (Figure 6H). No differences were observed among groups when evaluated according to the treatment administered. Regardless of the staining used, HE staining was more effective in showing the infiltration of immune cells and haemocytes (H) at the site of infection, especially in regions with larger yeast clusters (L). These regions were characterised by intense melanisation (M) involving yeasts and granuloma formation. In addition, budding yeasts, tissue invasion, and necrotic tissue were observed, indicating severe infection (Figure 5A and Figure 6C). PAS staining further defined the yeast morphology and aggregates, revealing several regions of melanisation (Figure 5B). Large aggregates of yeast were found within the granulomas, suggesting biofilm formation.

The difference among treatments was quantified as the number of granulomas found along the histopathological section (Figure 9). Overall, NEA treatment resulted in a lower number of granulomas compared to the AmB-treated group. On the other hand, depending on the species, a higher number of granulomas were observed in InP3 and SP96, both with more than 100 granulomas in the whole tissue.

## 4. Discussion

*C. auris* is an emerging yeast of worldwide interest due to its antifungal resistance and mortality rates due to its potential ability to cause systemic infections with a high mortality rate. To provide new treatment alternatives against *C. auris* infections, we have developed and evaluated both in vitro and in vivo a nanoemulsion loaded with AmB. In our study, NEA is more effective against planktonic cells than AmB, showing low MIC_50_ and MIC_90_ values for clades II, III, and IV (JAP 1, SP96, and VEN C6). This contrasts with the results reported by Marena et al. [14], who found no MIC differences between NEA and AmB against *C. auris* CDC B11903.

Fungal biofilms represent a serious health problem, particularly in the hospital environment, because their presence makes it difficult for antifungal drugs to penetrate the polymeric matrix (EPS) and achieve effective therapeutic activity [20,21]. Therefore, the identification of mechanisms or substances that inhibit biofilm formation is important for controlling infection and microbial resistance.

During biofilm formation, yeast produces EPS in the phases of surface attachment, cell recognition, and proliferation. In the beginning of formation, the percentage of EPS is low, making the yeast susceptible to the external environment, the host immune system, and the activity of antifungal drugs. [15,22,23]. In our study, we obtain high AmB activity against *C. auris* biofilm from Japan (JAP 1) and Spain (SP96) strains, which could be related to the lower presence of EPS, improving the inhibition of yeast growth by the drug. It is known that the East Asian strains, such as the Japanese strain (JAP 1) used in this study, have some peculiarities, such as increased susceptibility to fluconazole and low biofilm production. Due to these characteristics, these strains rarely cause systemic infections or large-scale outbreaks [24]. In our study, NEA inhibits biofilm formation by strains from India (InP13) and Venezuela (VEN C6), especially at lower concentrations. Similarly, a previous study evaluating the activity of NEA against *C. auris* strain CDC B11903 showed that NEA exhibited better inhibition of biofilm formation compared to AmB at lower concentrations (0.02 to 0.19 µg/mL) [15]. However, this study had the limitation of containing only one clinical isolate from each clade. To confirm these results, a large-scale study is needed.

It has been proven that nanoscale systems, such as NE, efficiently inhibit biofilms because nanoparticles are more likely to penetrate the EPS matrix at different stages of formation [23]. In our study, when mature biofilms of *C. auris* were treated with NEA and AmB, the biofilm structure was destabilised and was more susceptible to drug activity, even at lower concentrations. Thus, possible toxicities and side effects of AmB can be minimised using lower drug concentrations.

The *G. mellonella* model provides a versatile tool for assessing various aspects of drug activity, such as acute toxicity, prophylactic potential, antifungal efficacy, immunological response, and haemocyte count [25,26]. In this context, the cellular immune response mediated by haemocytes and the humoral response mediated by effector molecules such as antimicrobial peptides, melanin, and opsonins can efficiently immobilise or kill pathogens [27]. In our study, the in vivo results obtained using *G. mellonella* are consistent with the in vitro findings, where NEA demonstrated significantly higher antifungal activity than AmB, with a reduction in the fungal load in the haemolymph and tissues.

Arias et al. [28] infected *G. mellonella* larvae with *C. auris* NCPF 8973 and NCPF 8978 strains to assess the antifungal activity of chitosan. Their results revealed a significant antifungal effect of chitosan, with a reduction in the fungal load of both strains. Additionally, chitosan increased the vitality of the infected *G. mellonella* without causing acute toxicity. Similarly, Marena et al. [14] demonstrated that NEA exhibited better antifungal activity in a group of *G. mellonella* infected with *C. auris* CDC B11903. The authors attributed the improved efficacy to the characteristics of the system, including the physical structure, enhanced bioavailability, improved selectivity to the pathogen (cholesterol interacts with the fungal wall ergosterol), protection of the drug from enzymatic and cellular attacks, and better penetration into the fungal membrane via fusion, passive transport, or electrostatic interaction. Moreover, the authors observed that NEA did not cause acute toxicity in *G. mellonella*. Therefore, in addition to demonstrating superior antifungal activity, NEA was considered safe as it did not show acute toxicity up to 1000 µg/mL in previous studies.

Furthermore, the *G. mellonella* model allows the quantification of the fungal load using histopathological analysis. Following fungal infection, the innate immune cell response results in the aggregation of haemocytes into various granular-like structures or cell clusters. These structures, known as granulomas, form during an inflammatory response to surround the pathogen and prevent its spread. These granulomas often contain large amounts of melanin, which has antimicrobial properties and can be observed micro- and macroscopically after infection. Necrotic areas, characterised by a dark orange colour, may also be present within the granulomas [29].

In our study, all clades of *C. auris* assayed produced a rapid progression of infection, with dissemination into tissues and organelles. Histological analysis revealed the recruitment of haemocytes and yeasts in granulomas with a high melanin index. In addition, most of these granulomas were more frequently located between the lower extremities (tail) and upper extremities (thorax and head). Invertebrates, including *G. mellonella,* have an open circulatory system that runs through the entire body of the larva, mainly concentrated in the central region (abdomen) where most of the haemolymph is present. The haemolymph has several vital functions, such as the transport of lipids and proteins and the neutralisation of endotoxins, in supporting the larvae and protecting against microbial infection. This distribution pattern of the circulatory system in *G. mellonella* may explain the development of granulomas in regions of reduced circulation, such as the extremities [30,31].

Similarly, Muñoz et al. [32] evaluated the virulence of two *C. auris* strains belonging to clade IV (Ca432 and Ca386) in a *G. mellonella* model and found an increased amount of yeast in the tissues, revealing an infection and infiltration profile without the presence of hyphae or pseudohyphae. In contrast, Garcia-Bustos et al. [19], who evaluated the virulence of ten bloodstream *C. auris* strains using the *G. mellonella* model, observed the spread of infection after 24 h inoculation, especially for non-aggregative strains. The dissemination in the haemolymph was most evident in the early stages of infection. The authors also noted the presence of pseudohyphae in the haemolymph and tissues.

## 5. Conclusions

It is imperative to develop new therapeutic methods for the management of *C. auris* infections due to the increasing number of cases and associated resistance. In our study, amphotericin B-loaded nanoemulsions show improved in vitro and in vivo antifungal activity against *C. auris*. They offer promising alternatives to control the infection. Similarly, the metabolic activity of mature biofilms is significantly reduced, indicating a higher efficacy of NEA treatment. This demonstrates that encapsulation enhances the therapeutic activity of AmB, which is also reflected in the control of infection in the in vivo model. Furthermore, the absence of differences between infection control group and AmB treatment group demonstrates the efficacy of encapsulation. Although studies with a larger number of isolates are required, nanoemulsions can be considered intelligent drug delivery systems that enhance the therapeutic activity of drugs such as AmB.

## Figures and Tables

**Figure 1 microorganisms-11-01626-f001:**
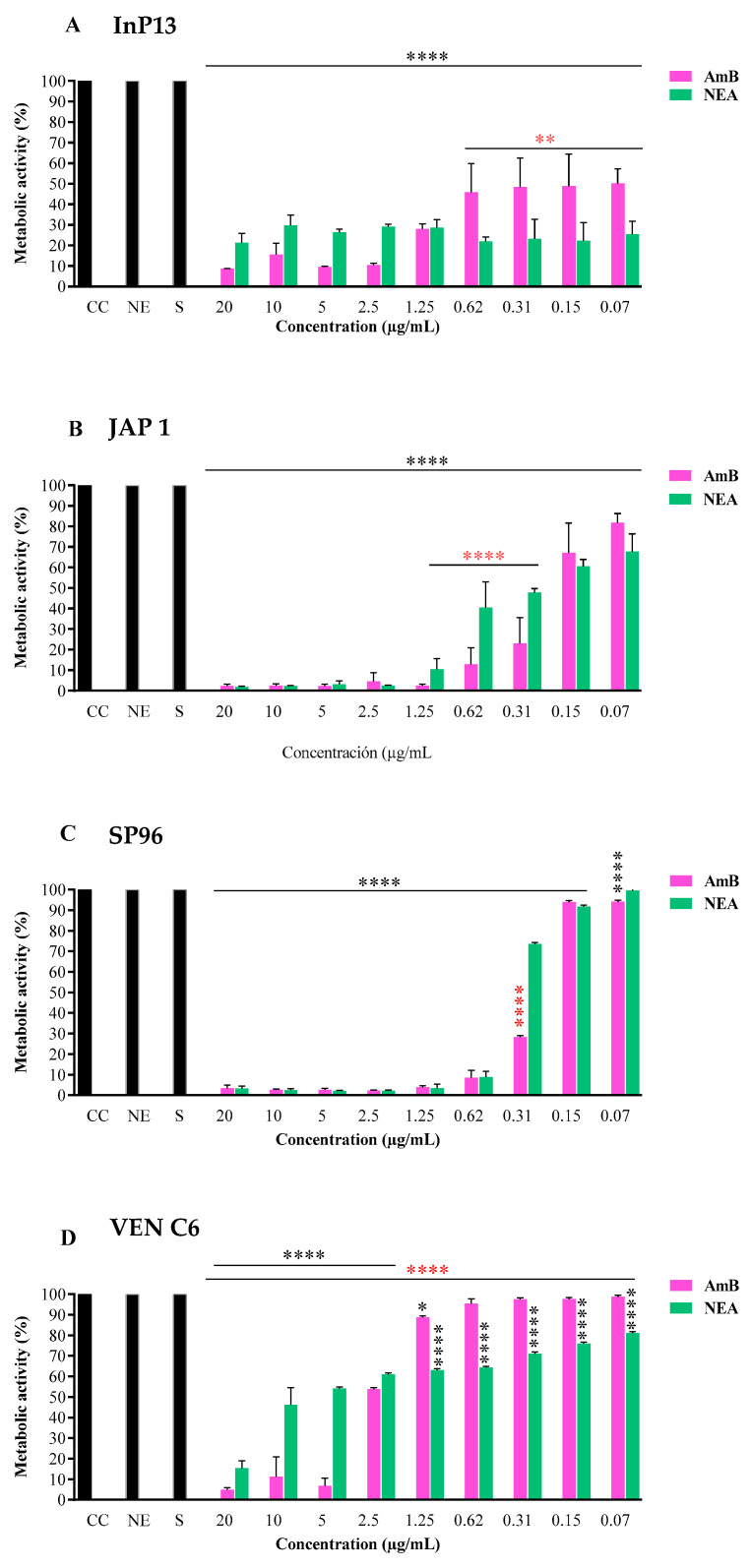
Activity of NEA and AmB against *C. auris* biofilm formation. NEA: nanoemulsion + amphotericin B; NE: nanoemulsion; S: solvent (DMSO 5% + YEPD); CC: growth control; AmB: amphotericin B. *p*-values < 0.0001 (****), *p* < 0.002 (**), and *p* < 0.01 (*). Black (****) indicates the difference between the treated group (AmB or NEA) and the control group (growth control, S, and NE). Red (****) indicates a statistically significant difference between treatments (NEA and AmB, respectively).

**Figure 2 microorganisms-11-01626-f002:**
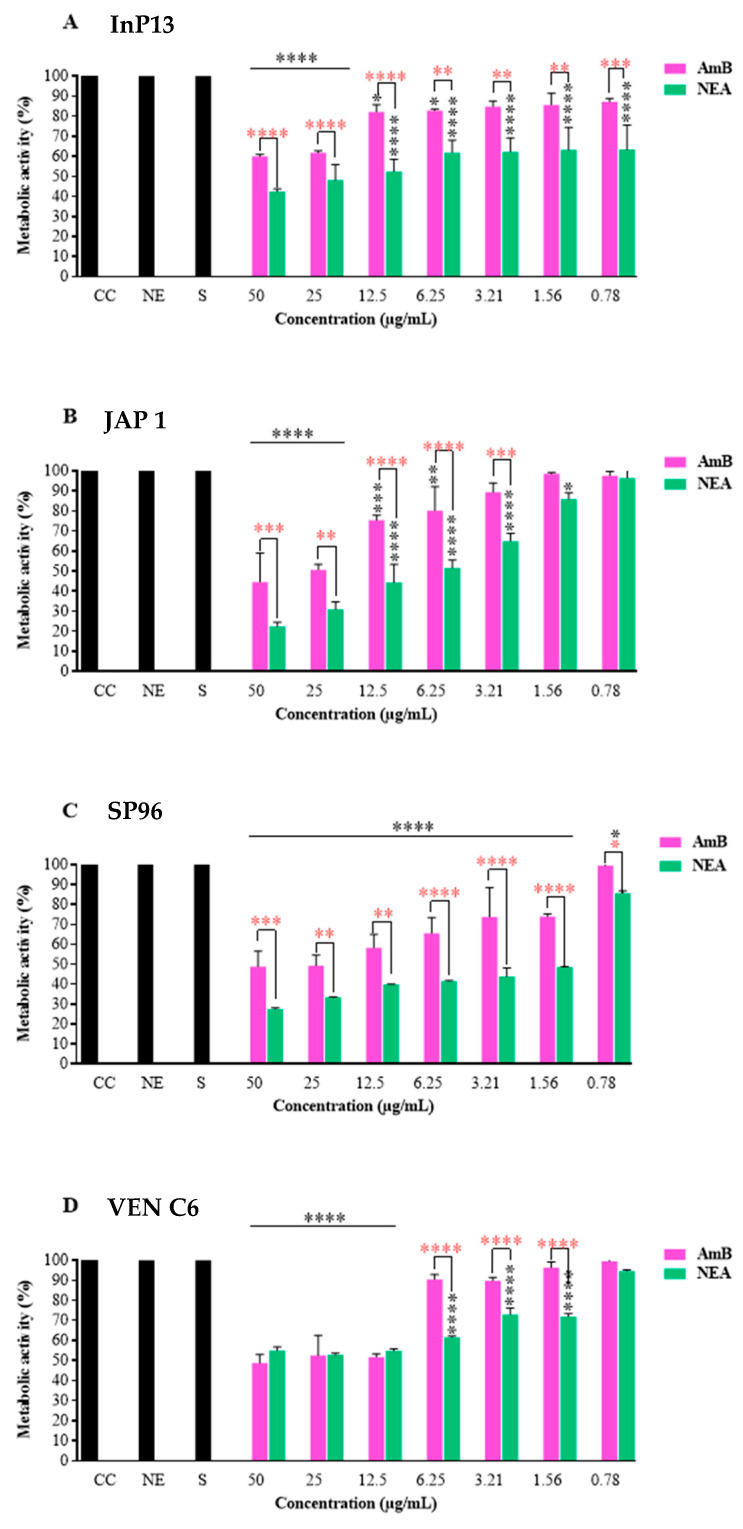
Activity of AmB and NEA in mature biofilms of *C. auris*. NEA: nanoemulsion + amphotericin B; NE: nanoemulsion; S: solvent (DMSO 5% + YEPD); CC: growth control; AmB: amphotericin B. *p*-values < 0.0001 (****), *p* < 0.003 (***), *p* < 0.002 (**), and *p* < 0.01 (*). Black (****) indicates the difference between the treated group (AmB or NEA) compared to the control group (growth control, S, and NE). Red (****) indicates a statistical difference between the treatments (NEA and AmB).

**Figure 3 microorganisms-11-01626-f003:**
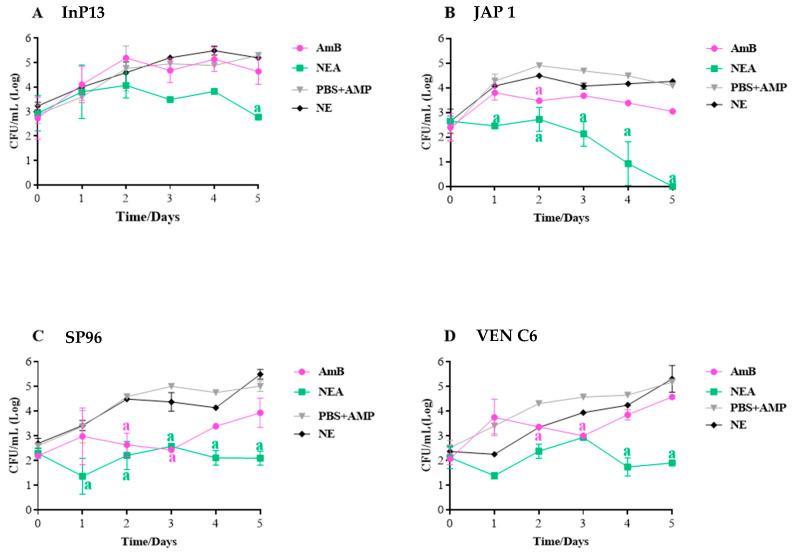
Antifungal activity of NEA and AmB in the haemolymph of *G. mellonella* infected with *C. auris*. NEA: nanoemulsion + amphotericin B; AmB: amphotericin B; PBS + AmP: phosphate-buffered saline + ampicillin (20 µg/mL) with infection; NE: nanoemulsion with infection. NE and PBS + AmP: considered control groups (without therapy); the presence of letters indicates statistical difference between the infection control group (PBS + AmP or NE) and the treated group (AmB and NEA). Time 0 indicates the fungal load after 2 h of infection.

**Figure 4 microorganisms-11-01626-f004:**
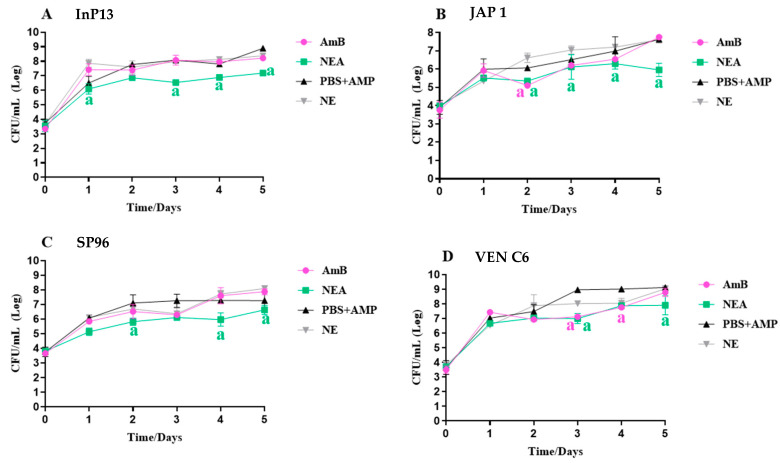
Quantification of the fungal load of *C. auris* in infected *G. mellonella* tissues treated with NEA and AmB. NEA: nanoemulsion + amphotericin B; AmB: amphotericin B; PBS + AmP: phosphate-buffered saline + ampicillin 20 µg/mL with infection; NE: nanoemulsion with infection; NE and PBS + AmP: considered control groups or untreated groups. The absence of letters indicates that there is no statistical difference; the presence of letters indicates a statistical difference between the infection control group (PBS + AmP or NE) and the treated group (AmB and NEA). Time 0 indicates the fungal load after 2 h of infection. n = 3 larva/time.

**Figure 5 microorganisms-11-01626-f005:**
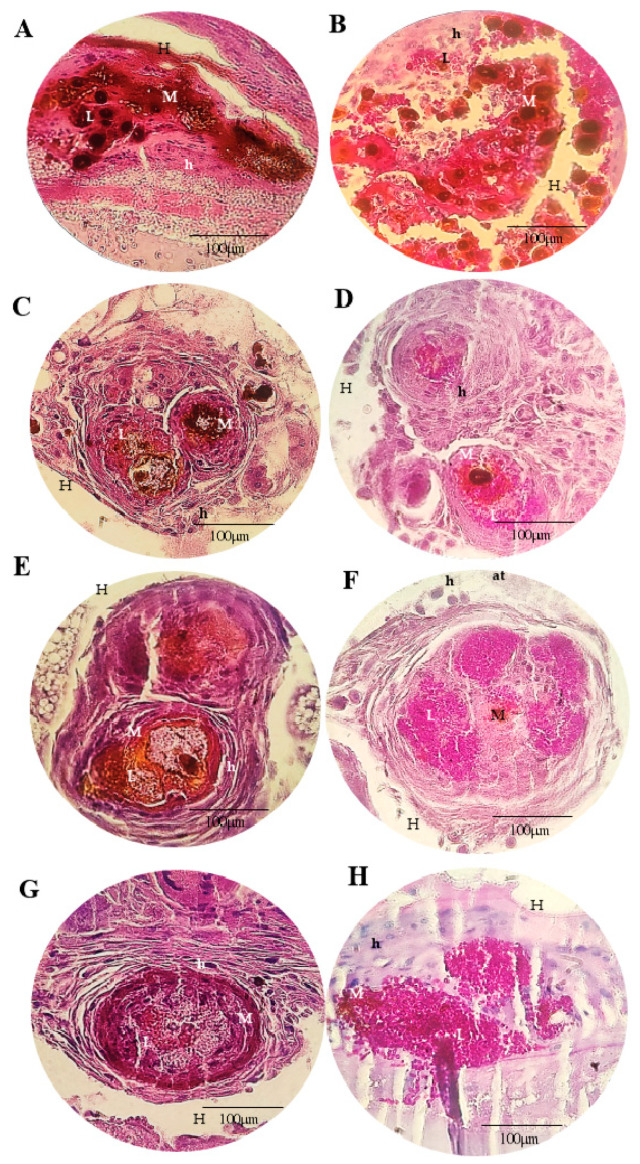
Histopathological findings of *G. mellonella* larvae infected with *C. auris* InP13 strain stained with HE (**A**,**C**,**E**,**G**) and PAS (**B**,**D**,**F**,**H**), 100× magnification. (**A**,**B**): PBS + ampicillin; (**C**,**D**): nanoemulsion; (**E**,**F**): amphotericin B; (**G**,**H**): nanoemulsion + amphotericin B; (M): melanisation; (L): cluster of yeasts; (at): adipose tissue; (H): haemolymph; (h): haemocytes.

**Figure 6 microorganisms-11-01626-f006:**
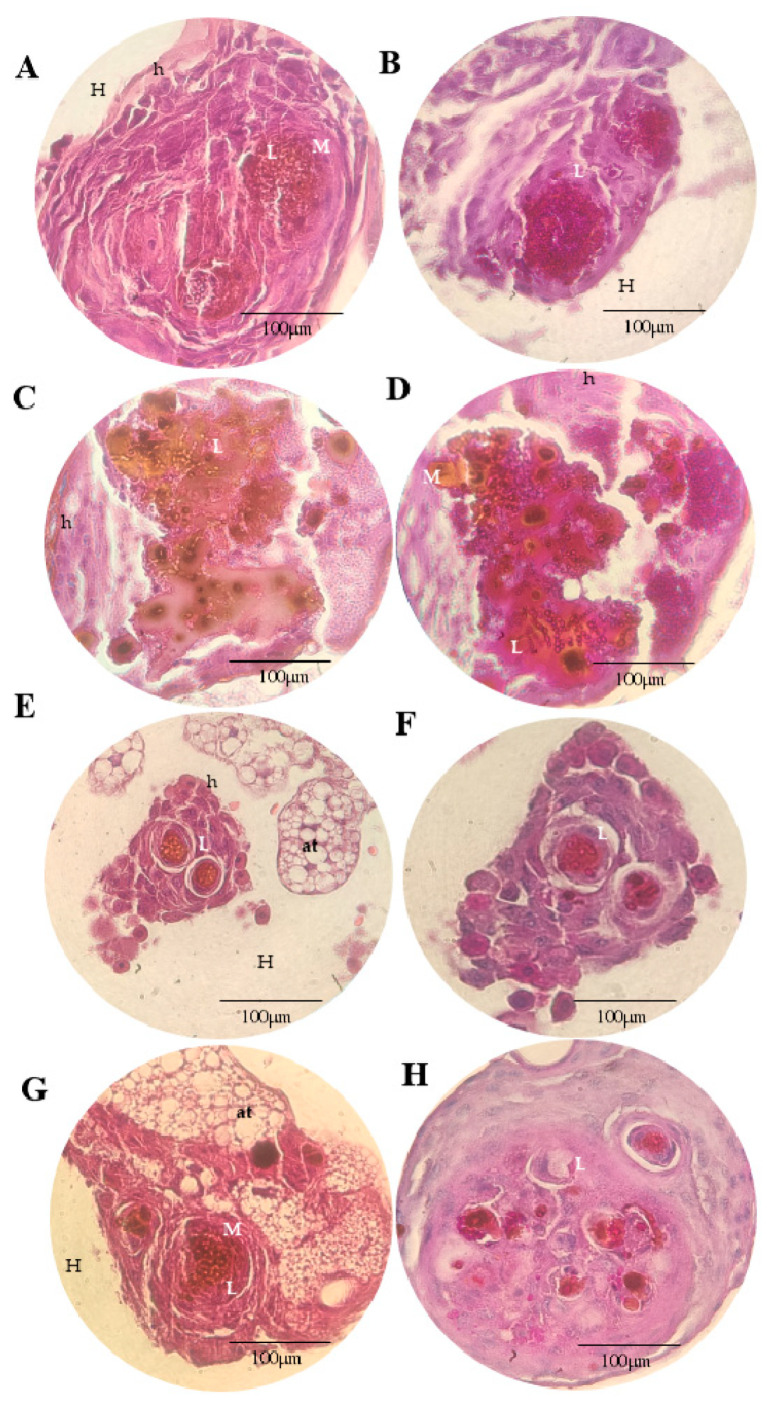
Histopathological findings of *G. mellonella* larvae infected with the *C. auris* JAP 1 strain stained with HE (**A**,**C**,**E**,**G**) and PAS (**B**,**D**,**F**,**H**), 100× magnification. (**A**,**B**): PBS + ampicillin; (**C**,**D**): nanoemulsion; (**E**,**F**): amphotericin B; (**G**,**H**): nanoemulsion + amphotericin B; (M): melanisation; (L): cluster of yeasts; (at): adipose tissue; (H): haemolymph; (h): haemocytes.

**Figure 7 microorganisms-11-01626-f007:**
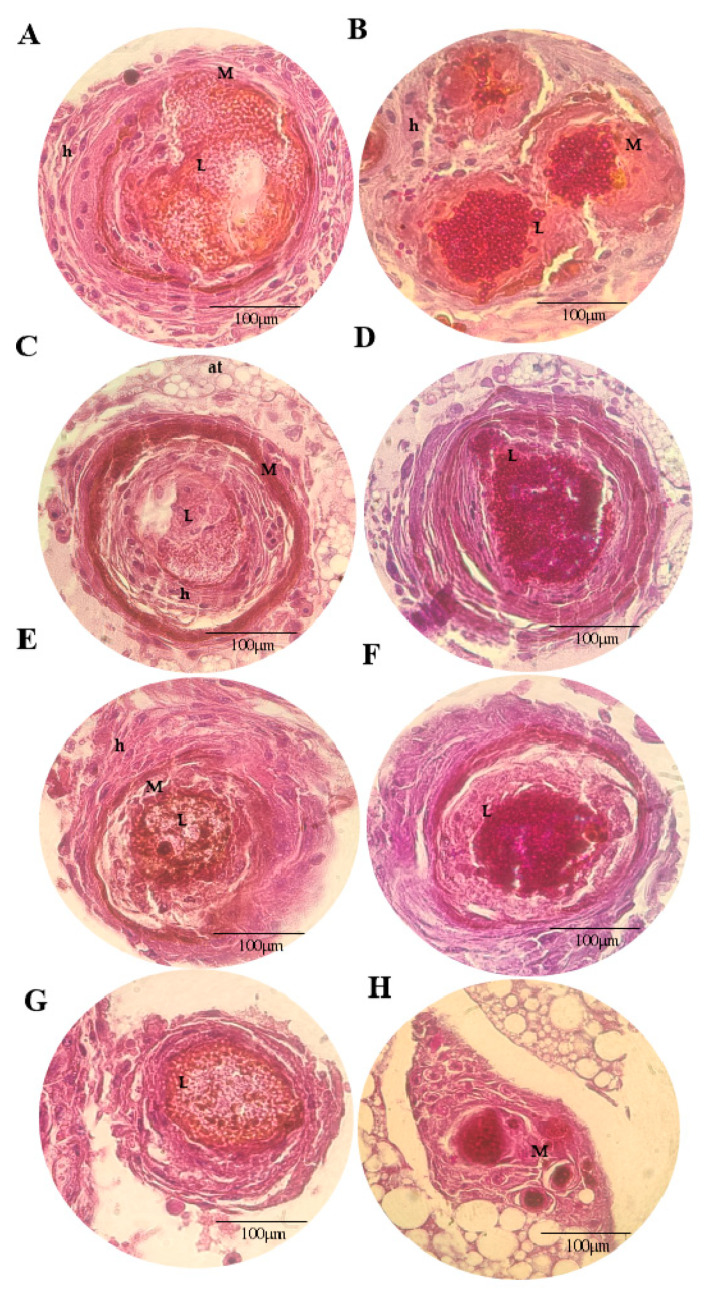
Histopathological findings of *G. mellonella* larvae infected with the *C. auris* SP96 strain, stained with HE (**A**,**C**,**E**,**G**) and PAS (**B**,**D**,**F**,**H**), 100× magnification. (**A**,**B**): PBS + ampicillin; (**C**,**D**): nanoemulsion; (**E**,**F**): amphotericin B; (**G**,**H**): nanoemulsion + amphotericin B; (M): melanisation; (L): cluster of yeasts; (at): adipose tissue; (H): haemolymph; (h): haemocytes.

**Figure 8 microorganisms-11-01626-f008:**
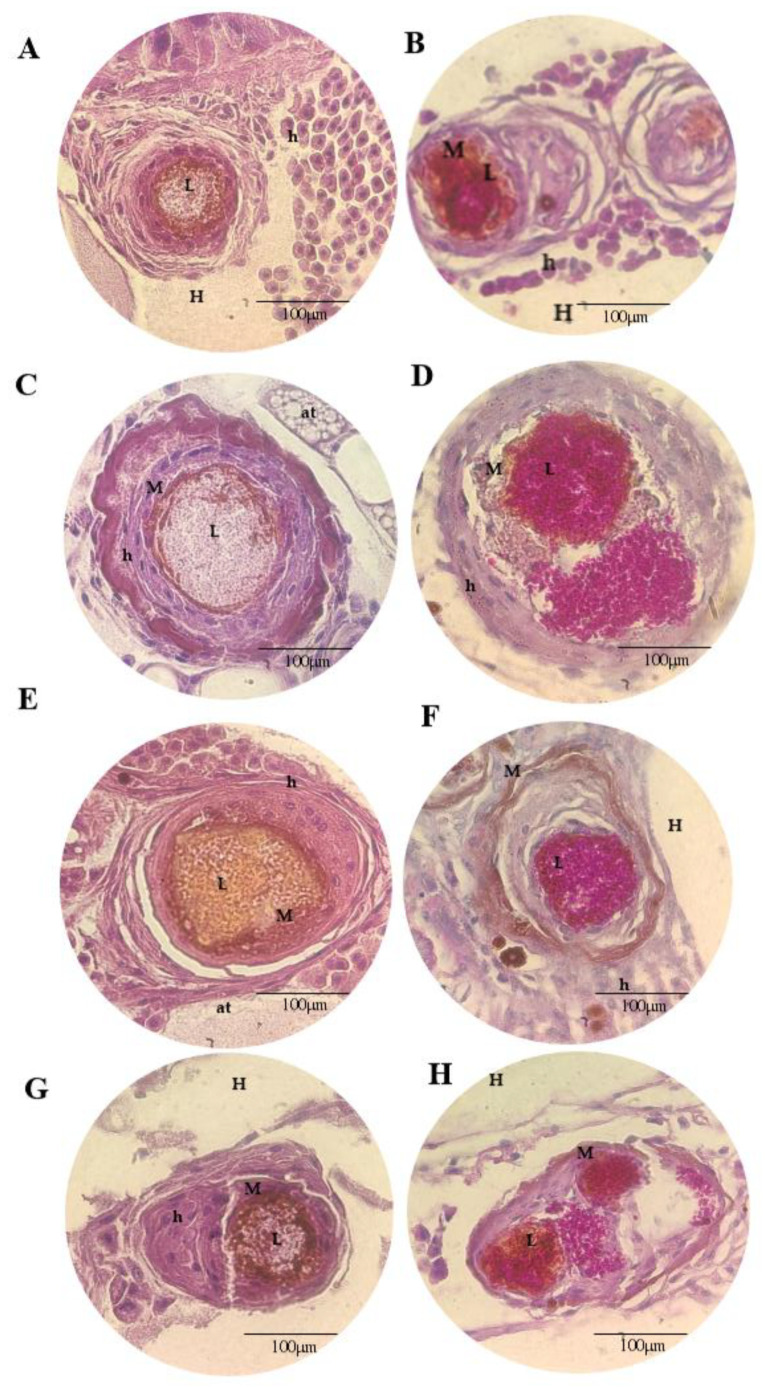
Histopathological findings of *G. mellonella* larvae infected with the *C. auris* VEN C6 strain stained with HE (**A**,**C**,**E**,**G**) and PAS (**B**,**D**,**F**,**H**), 100× magnification. (**A**,**B**): PBS + ampicillin; (**C**,**D**): nanoemulsion; (**E**,**F**): amphotericin B; (**G**,**H**): nanoemulsion + amphotericin B; (M): melanisation; (L): cluster of yeast; (at): adipose tissue; (H): haemolymph; (h): haemocytes; (P): budding or pseudohyphae.

**Figure 9 microorganisms-11-01626-f009:**
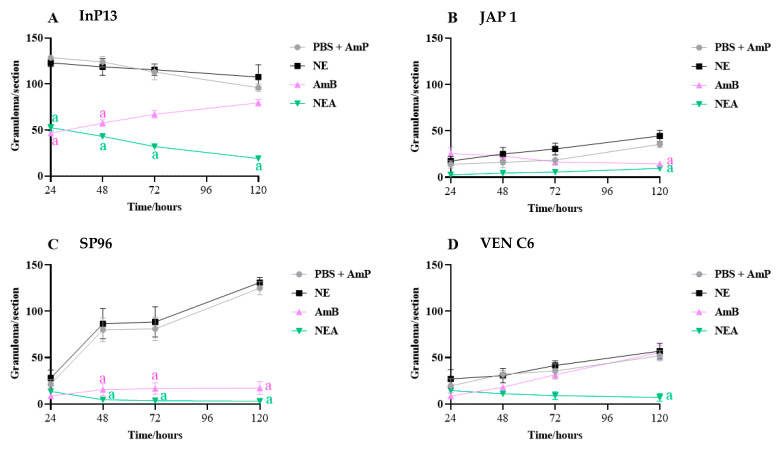
Total count of granulomas in histopathological sections of *G. mellonella* treated with NEA and AmB. AmP: ampicillin; NE: nanoemulsion; AmB: amphotericin B; NEA: nanoemulsion + amphotericin B; NE and PBS + AmP: considered control groups or group without therapy. The absence of a letter means that there is no statistical difference; the presence of letters indicates a statistical difference compared with the control groups (PBS + AmP and NE); n = 3 larva/time.

**Table 1 microorganisms-11-01626-t001:** Minimum inhibitory concentration of AmB and NEA in *C. auris*.

	Minimum Inhibitory Concentration (µg/mL) *Candida auris*							
Strains/ Treatment	InP13		JAP 1		SP96		VEN C6	
	MIC_90_	MIC_50_	MIC_90_	MIC_50_	MIC_90_	MIC_50_	MIC_90_	MIC_50_
AmB	0.62 ± 0.0	0.41 ± 0.1	0.31 ± 0.18	0.1308 ± 0.10	0.62 ± 0.0	0.137 ± 0.05	1.25 ± 0.22	0.325 ± 0.222
NEA	1.25 ± 0.0	0.029 ± 0.002	0.038 ± 0.02	0.004 ± 0.004	0.038 ± 0.06	0.002 ± 0.001	0.31 ± 0.0	0.003 ± 0.001

AmB: amphotericin B; NEA: nanoemulsion + amphotericin B; MIC_90_: minimum inhibitory concentration 90%; MIC_50_: minimum inhibitory concentration 50%. Standard deviation (±SD).

## Data Availability

The data presented in this study are available on request from the corresponding author.

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
