# Peer review of "Nanoemulsion Increases the Antifungal Activity of Amphotericin B against Four Candida auris Clades: In Vitro and In Vivo Assays"

_microorganisms, 2023, doi:10.3390/microorganisms11071626_

Round 1

Reviewer 1 Report

microorganisms-2339176 Nanoemulsion increases the antifungal activity of amphotericin B against four Candida auris clades: in vitro and in vivo assays Authors described the anti-Candida aurisis of a nanoemulsion with amphotericin B (NEA) against planktonic cells and biofilm of clinical isolates of C. auris belonging to four different clades. The topic is interesting, however in presented manuscript there is lack of several information. In present version it is very chaotic and difficult to read. Please read carefully your paper and put some effort to organise it. Firstly, there are several lacks in Methodology section, there is no proper description of controls. More warring is lack of data from untreated G. mellonella controls. There is no information about the origin of strains. Moreover not all methodology is described. Next, in result section in some points, there is lack of proper description of data, also the presentation of histopathological results is incomprehensible. There is lack of discussion section, I mean there are some short fragments in result section which seems to be an attemp of discussion, however presented research require a proper discussion. Despite these, I think the topic is very interesting and if authors put in right order some elements it can be a valuable publication. Minor comments L72 - Please add the hypothesis L79 - What was the origin of strains? L90 - Please add information about the controls (solvent control, cells treatment only with AmB and cells treatment only with NEA) L99 - Describe the modification in details L112/L117 - How many repetition did you make and add information about the controls (solvent control) L122 - Put your information about the development stage of larvae and the orgin/methodology of insect rearing. Also information about controls are needed (untreated insect ) L134 - Please put more detail about the hemolimph collection L142  -Please check the normality of the data Table 1 - Please add the statistic analysis ( the SD, mark the statistically important differences, add information about the statistical test) L168 - L176 - This is more introduction fragment, not suitable in result section. L244 - Add information about the statistical test Figure 4a - There is a statistical difference marked as "b", but is compared with what? L267-L287 - It is more introduction/discussion, does not relevant in result section L297 - There is no information about this experiments in Methodology Figure 5 - lack of scal bar and low size of figures 

Author Response

Manuscript ID: microorganisms-2339176

Title: Nanoemulsion increases the antifungal activity of amphotericin B against four Candida auris clades: in vitro and in vivo assays

Dear reviewer

Journal Microorganisms

We are sending an article entitled “Nanoemulsion increases the antifungal activity of amphotericin B against four Candida auris clades: in vitro and in vivo assays” authored by Gabriel Davi Marena, Alba Ruiz-Gaitán, Victor Garcia-Bustos, Maria Ángeles Tormo-Mas, Jose Manuel Perez-Royo, Alejandro Lopez, Patricia Bernabe, María Dolores Pérez Ruiz, Lara Zaragoza Macian, Carmen Vicente Saez, Antonia Avalos Mansilla, Eulogio Valentín Gómez , Gabriela Correa Carvalho, Tais Maria Bauab, Marlus Chorilli and Javier Pemán.

In view of the increase in infectious diseases, microbial resistance, and the emergence of new lethal species, this manuscript presents important results of pharmaceutical nanotechnology in the development of nanoemulsions loaded with amphotericin B for the treatment of Candida auris, a new resistant species.

The manuscript has been completely rewritten to make this easier to understand. For this reason, we omitted the change control because the totality of the article is different. We hope that this revision improves the quality of the manuscript for publication. This manuscript presents promising and encouraging results since the nanoemulsions loaded with amphotericin B showed better antifungal activity against planktonic cells, biofilms, and infections in an alternative in vivo model of Galleria mellonella.

Yours sincerely,

Dra. Alba Ruiz-Gaitán MD, PhD

On behalf of all the authors of the article referred to above

Review 1

1) microorganisms-2339176 Nanoemulsion increases the antifungal activity of amphotericin B against four Candida auris clades: in vitro and in vivo assays Authors described the anti-Candida aurisis of a nanoemulsion with amphotericin B (NEA) against planktonic cells and biofilm of clinical isolates of C. auris belonging to four different clades. The topic is interesting, however in presented manuscript there is lack of several information. In present version it is very chaotic and difficult to read. Please read carefully your paper and put some effort to organise it. Firstly, there are several lacks in Methodology section, there is no proper description of controls.

Thank you for reviewing the manuscript. The manuscript has been completely rewritten and the length reduced according to the recommendations of the reviewers.

2) More warring is lack of data from untreated G. mellonella controls.

Dear reviewer, thank you for reviewing the manuscript. The description of this section has been added to the text and is as follows:

2.4. Antifungal assay using an in vivo model of Galleria mellonella

The day before infection, healthy cream-colored G. mellonella larvae of 250-350 mg (Dnatecosistemas, Madrid, Spain) were placed in a plastic container with ventilation holes and incubated at 37 °C for adaptation. On the day of infection, the larvae were kept in a Petri dish with ice for 2 - 3 min for immobilization (groups of 20 larvae). Prior to inoculation, the larva prolegs were disinfected with a 70% alcohol-impregnated swab. Then, using a Hamilton Microliter™ syringe, 10 µL of C. auris inoculum (104 cells/larva) resuspended in PBS + AmP (Ampicillin, Sigma Aldrich, Steinheim, North Rhine-Westphalia, Germany) at a concentration of 20 µg/mL was administered into the penultimate proleg of each larva [18]. Larvae were incubated at 37 °C for 2 h. AmB was solubilized in PBS + AmP + DMSO (0.05%), and NEA was solubilized in PBS + AmP. After 2h, 10µL of AmB and NEA (1.68 mg/kg - 3x MIC) was injected into the penultimate proleg. The treatment was carried out every day for 5 days. Controls for the G. mellonella in vivo assay were as follows: i) the group without treatment and without infection (negative group), ii) the group infected and treated with PBS +AmP (diluent control), and iii) the group infected and treated with NE without drug (to confirm that NE has no antifungal activity). All the larvae were incubated at 37 °C for 24 h. The syringe was immersed in 70% ethanol for 2-3 min, rinsed five times with 70% ethanol, and then rinsed five times with sterile 1× PBS between the strain procedures.

3) There is no information about the origin of strains.

The origin of strains InP13, JAP 1, SP96 and VEN C6 are from India, Japan, Spain, and Venezuela, respectively. Information added to the manuscript.

4) Moreover not all methodology is described.

Dear reviewer, we added new information to the methodology of the manuscript.

5) Next, in result section in some points, there is lack of proper description of data, also the presentation of histopathological results is incomprehensible.

Dear reviewer, the results section has been completely rewritten and separated from the discussion section to make it easier to read.

6) There is lack of discussion section, I mean there are some short fragments in result section which seems to be an attemp of discussion, however presented research require a proper discussion.

Dear reviewer, the results section has been completely rewritten and separated from the discussion section to make it easier to read.

 7) Minor comments L72 – Please add the hypothesis

 Dear reviewer, we added the hypothesis to the manuscript.

“NE is expected to improve the antifungal activity of AmB against the different clades of C. auris both in biofilms and in vivo studies of G. mellonella

8)  L79 – What was the origin of strains?

Dear reviewer, the information has been added and is as follows:

“The antifungal activity of AmB and NEA were evaluated against four clinical isolates of C. auris belonging to four different clades: C. auris VPCI479/P13 (India, CLADE I – InP13), C. auris CBS10913 (Japan, CLADE II – JAP 1), C. auris CBS 15603 (Spain, CLADE III – SP96), and C. auris VEN C6 (Venezuelan, CLADE IV)”.

9)  L90 – Please add information about the controls (solvent control, cells treatment only with AmB and cells treatment only with NEA)

We added to the manuscript.

The controls for the minimum inhibitory determination (item 2.2) were NE without AmB + inoculum (proving that NE has no antifungal action); solvent = RPMI + 5% DMSO + inoculum (check that the solvent has no antifungal action); growth control (inoculum + RPMI); RPMI sterility control; Sterility control of AmB and NEA (Treatment + RPMI.

10) L99 - Describe the modification in detail

We added to the manuscript.

The modifications were: “the use of 96-well microplates, Yeast Extract Peptone Dextrose culture medium, 2 h for pre-adhesion without rotation, incubation for up to 48 h and washing with PBS”.

 11) L112/L117 - How many repetition did you make and add information about the controls (solvent control)

We added this information to the manuscript.

The experiment was carried out at least three times and on three different days.

 L122 - Put your information about the development stage of larvae and the origin/methodology of insect rearing. Also, information about controls are needed (untreated insect)

Dear reviewer, the larvae were obtained by the distributor Dnatecosistema from Madrid, Spain. All larvae arrived ready for testing. We added this information to the manuscript.

 12) L134 - Please put more detail about the hemolimph collection

We added to the manuscript.

  Section 2.4: “after decapitating the larvae, they were trapped with the thumb and index finger of the hand and turned upside down so that all the hemolymph left the larval body into the Eppendorf”

13) L142 -Please check the normality of the data Table 1 - Please add the statistical analysis (the SD, mark the statistically important differences, add information about the statistical test)

We added to the manuscript.

14)  L168 - L176 - This is more introduction fragment, not suitable in result section.

Dear reviewer, we removed this paragraph from the results.

15) L244 - Add information about the statistical test Figure 4a - There is a statistical difference marked as "b", but is compared with what?

The letter "b" indicates that NEA is also different from AmB (with AmB different from the control group). We added this information to the manuscript.

16) L267-L287 - It is more introduction/discussion, does not relevant in result section

Dear reviewer, we have completely rewritten the Results and Discussion sections.  They have been separated to make the manuscript more understandable.

17)  L297 - There is no information about this experiments in Methodology Figure 5 - lack of scal bar and low size of figures.

Dear reviewer, we added the histopathology methodology, item 2.5.

Added scaling and improved image formatting, item 3.2.4.

Reviewer 2 Report

The study aimed to analyze the in vitro and in vivo antifungal activity of a nanoemulsion with amphotericin B (NEA) against planktonic cells and biofilm of clinical isolates of C. auris belonging to four different clades is very interesting, providing new and valuable information on better therapeutic options for patients.

Only few suggestions to improve the manuscript:

1. Brief manifestation of the limitations of the study, basically refering to the number of isolates used to test the nanoemulsion activity (only 4 isolates is a very reduced number)

2. Since the discussion is immersed within each result section, conclusions in point 4 could be more detailed in regards to the main results

3. Figures, specially in the histopathology section, are suggested to be reduced in number

Author Response

Manuscript ID: microorganisms-2339176

Title: Nanoemulsion increases the antifungal activity of amphotericin B against four Candida auris clades: in vitro and in vivo assays

Dear reviewer,

Journal Microorganisms

We are sending an article entitled “Nanoemulsion increases the antifungal activity of amphotericin B against four Candida auris clades: in vitro and in vivo assays” authored by Gabriel Davi Marena, Alba Ruiz-Gaitán, Victor Garcia-Bustos, Maria Ángeles Tormo-Mas, Jose Manuel Perez-Royo, Alejandro Lopez, Patricia Bernabe, María Dolores Pérez Ruiz, Lara Zaragoza Macian, Carmen Vicente Saez, Antonia Avalos Mansilla, Eulogio Valentín Gómez , Gabriela Correa Carvalho, Tais Maria Bauab, Marlus Chorilli and Javier Pemán.

In view of the increase in infectious diseases, microbial resistance, and the emergence of new lethal species, this manuscript presents important results of pharmaceutical nanotechnology in the development of nanoemulsions loaded with amphotericin B for the treatment of Candida auris, a new resistant species.

The manuscript has been completely rewritten to make this easier to understand. For this reason, we omitted the change control because the totality of the article is different. We hope that this revision improves the quality of the manuscript for publication. This manuscript presents promising and encouraging results since the nanoemulsions loaded with amphotericin B showed better antifungal activity against planktonic cells, biofilms, and infections in an alternative in vivo model of Galleria mellonella.

Yours sincerely,

Dra. Alba Ruiz-Gaitán MD, PhD

On behalf of all the authors of the article referred to above

Review 2.

The study aimed to analyze the in vitro and in vivo antifungal activity of a nanoemulsion with amphotericin B (NEA) against planktonic cells and biofilm of clinical isolates of C. auris belonging to four different clades is very interesting, providing new and valuable information on better therapeutic options for patients.

Only few suggestions to improve the manuscript:

1) Brief manifestation of the limitations of the study, basically refering to the number of isolates used to test the nanoemulsion activity (only 4 isolates is a very reduced number)

Dear reviewer, due to in vivo assays, we decided to work with only 1 strain of each clade. However, we agree that this low sample of fungal strain is a limiting factor in this study, and we added this information to the manuscript

2) Since the discussion is immersed within each result section, conclusions in point 4 could be more detailed in regards to the main results

Dear reviewer, we have added this information to the conclusion:

“ It is imperative to develop new therapeutic methods for the management of C. auris infections due to the increasing number of cases and associated resistance. In our study, amphotericin B-loaded nanoemulsions show improved in vitro and in vivo antifungal activity against C. auris. They offer promising alternatives to control the infection. Similarly, the metabolic activity of mature biofilms is significantly reduced, indicating a higher efficacy of NEA treatment. This demonstrates that encapsulation enhances the therapeutic activity of AmB, which is also reflected in the control of infection in the in vivo model.  Furthermore, the absence of differences between infection control group and AmB treatment group demonstrates the efficacy of encapsulation. Although studies with a larger number of isolates are required, nanoemulsions can be considered as intelligent drug delivery systems that enhance the therapeutic activity of drugs as AmB”.

3) Figures, specially in the histopathology section, are suggested to be reduced in number

Dear reviewer, as there was no difference in granuloma formation between days, we selected the best images from the last day of infection to better organize and discuss the data. The new images are shown in Figures 5 to 8.

Reviewer 3 Report

*Comments For the authors*:

The authors of the MS entitled “Nanoemulsion increases the antifungal activity of amphotericin B against four /Candida auris /clades: /in vitro /and /in vivo /assays, report MIC and in vivo data of an amphotericin B nanoemulsion against 4 isolates of this species.

The data are abundant. But the MS needs to be shortened (Figures especially) to illustrate the best data for each of the parameters evaluated, instead of presenting all the data.

Some specific comments:

Fungal strains: provide the number of isolates and the highest and lowest MIC_90

MIC determination. The CLSI method is well known, just describe the relevant modifications.

2.4.1-2.4.4. The same comment as above, for all the sections.

Table 1. a little difficult to assess the data for the 4 isolates/clades; perhaps MIC_50 s and MIC_90s could be next to each other? That way you could easily see that 1 has the best in vitro

activity?

Fig. 2 is difficult to read/interpret, that is one of the most important results. T

The same applies to the graphical presentation of the histopathological evaluation data.

All those results need to be presented showing the best and the lowest only in an easy interpretation for the reader.

Author Response

Manuscript ID: microorganisms-2339176

Title: Nanoemulsion increases the antifungal activity of amphotericin B against four Candida auris clades: in vitro and in vivo assays

Dear reviewer,

Journal Microorganisms

We are sending an article entitled “Nanoemulsion increases the antifungal activity of amphotericin B against four Candida auris clades: in vitro and in vivo assays” authored by Gabriel Davi Marena, Alba Ruiz-Gaitán, Victor Garcia-Bustos, Maria Ángeles Tormo-Mas, Jose Manuel Perez-Royo, Alejandro Lopez, Patricia Bernabe, María Dolores Pérez Ruiz, Lara Zaragoza Macian, Carmen Vicente Saez, Antonia Avalos Mansilla, Eulogio Valentín Gómez , Gabriela Correa Carvalho, Tais Maria Bauab, Marlus Chorilli and Javier Pemán.

In view of the increase in infectious diseases, microbial resistance, and the emergence of new lethal species, this manuscript presents important results of pharmaceutical nanotechnology in the development of nanoemulsions loaded with amphotericin B for the treatment of Candida auris, a new resistant species.

The manuscript has been completely rewritten to make this easier to understand. For this reason, we omitted the change control because the totality of the article is different. We hope that this revision improves the quality of the manuscript for publication. This manuscript presents promising and encouraging results since the nanoemulsions loaded with amphotericin B showed better antifungal activity against planktonic cells, biofilms, and infections in an alternative in vivo model of Galleria mellonella.

Yours sincerely,

Dra. Alba Ruiz-Gaitán MD, PhD

On behalf of all the authors of the article referred to above

Review 3.

The authors of the MS entitled “Nanoemulsion increases the antifungal activity of amphotericin B against four /Candida auris /clades: /in vitro /and /in vivo /assays, report MIC and in vivo data of an amphotericin B nanoemulsion against 4 isolates of this species.

1) The data are abundant. But the MS needs to be shortened (Figures especially) to illustrate the best data for each of the parameters evaluated, instead of presenting all the data.

Dear reviewer, thank you very much for reviewing the manuscript. His suggestions will be very important for this study.

 As there was no difference in granuloma formation between days, we selected the best images from the last day of infection to better structure and discuss the data. The new images are shown in Figures 5 to 8, section 3.2.3.

2) Some specific comments:

Fungal strains: provide the number of isolates and the highest and lowest MIC_90

Dear reviewer, we have completely rewritten this paragraph to read as follows:

“3.1 Determination of the Minimum Inhibitory Concentration (MIC)

AmB and NEA MICs are shown in Table 1. The MIC90 of NEA was considerably lower than that of AmB for VEN C6 (MIC90 1.25 vs 0.31 µg/mL), JAP 1 (MIC90 0.31 vs 0.038 µg/mL) and SP96 (MIC90 0.62 vs 0.038 µg/mL). Additionally, NEA MIC50 values were lower. No fungal growth was observed in any of the control groups.”

3) MIC determination. The CLSI method is well known, just describe the relevant modifications.

Dear reviewer, we have completely rewritten.

4) 2.4.1-2.4.4. The same comment as above, for all the sections.

Dear reviewer, we removed some information, however, part of the methodology was maintained for better understanding. Furthermore, other reviewers suggest that the methodology be more detailed.

5) Table 1. a little difficult to assess the data for the 4 isolates/clades; perhaps MIC_50 s and MIC_90s could be next to each other? That way you could easily see that 1 has the best in vitro activity?

Dear reviewer, we have completely rewritten this section.

6) Fig. 2 is difficult to read/interpret, that is one of the most important results.Dear reviewer, we better organized the statistical results in Figure 2. The black asterisks indicate the difference between the treated group (AmB or NEA) compared to the control group (Growth Control, Solvent and NE). Red asterisks indicate differences between AmB and NEA, that is, between the treated group.

7) The same applies to the graphical presentation of the histopathological evaluation data. All those results need to be presented showing the best and the lowest only in an easy interpretation for the reader.

Dear reviewer, as there was no difference in granuloma formation between days, we selected the best images from the last day of infection to better structure and discuss the data. The new images are shown in Figures 5 to 8.

Reviewer 4 Report

This study aimed to analyze the in vitro and in vivo antifungal activity of a nanoemulsion with amphotericin B (NE) against planktonic cells and biofilm of clinical isolates of C. auris belonging to four different clades.  Overall, the article contains experimental findings that can be interesting for readers of Microorganisms. The authors should address some issues to improve the manuscript.

Page 2, lines 50-51: Please, check 106 cells/hour. I believe the correct is 10^6 cells/h. Hour is time unit and represented by h. Please change hour to h here and throughout the manuscript.

Page 3, lines 104-105: Was the supernatant removed and treated? 

Page 3, line 196: AmP is not defined. Please, check it.

The methodology for histology of the larvae was not provided. Authors should also add methodology for granuloma quantification.

I did not understand the statistical analysis reported by authors. Anova two way should be used to make comparisons within-groups over time and among groups at each experimental period. 

Page 4, lines 157-158: In methodology, authors report that MIC was determined by inactivation of 90% of yests. However, they provide MIC50. Regarding MIC50, all strains show lower MIC for NE than AMB. Please, verify.

Page 5, fig 1. Please, check y-axis of fig. 1B. Viability should be replaced by metabolic activity.

Pages 5-6, lines 190-200: Here, authors evaluated biofilm formation. Differences among strains were attributted to the presence of EPS but I am not sure if EPS could explain these data. In methodoly, authors described that non-adhered cells were treated. Is there any reference to support the lower EPS presence for JAP 1 and SP96 in biofilm formation? Indeed, the reference 22 should be used to discuss mature biofilms. 

Pages 8, fig.3: Authors should normalize the data. For InP13 (fig. 3A) this is ok, but for other strains there are some differences in the initial infection that may lead to a misinterpretation of the data. In addition, I was not able to understand the letters a and b. The legend is not complete.

Page 9, fig. 4: Once again, I did not understand the letters a and b. What do the statistically significant differences refer to?

Page 10, line 299: Were immune cells labeled "h" as hematocytes?

Figs 5-12: Please, add "original" before magnification. The micrographs should contain a bar since the magnification changes according to the image size. PAS was not defined (figs. 9-12).

Fig 13: The data should contain standard deviation. Please, add the number of samples in the legend. Indeed, the number of samples should be added in all figure legends of quantitative data.

Please, revise the text carefully. There are some typos (e.g., ESP instead EPS).

Author Response

Manuscript ID: microorganisms-2339176

Title: Nanoemulsion increases the antifungal activity of amphotericin B against four Candida auris clades: in vitro and in vivo assays

Dear reviewer,

Journal Microorganisms

We are sending an article entitled “Nanoemulsion increases the antifungal activity of amphotericin B against four Candida auris clades: in vitro and in vivo assays” authored by Gabriel Davi Marena, Alba Ruiz-Gaitán, Victor Garcia-Bustos, Maria Ángeles Tormo-Mas, Jose Manuel Perez-Royo, Alejandro Lopez, Patricia Bernabe, María Dolores Pérez Ruiz, Lara Zaragoza Macian, Carmen Vicente Saez, Antonia Avalos Mansilla, Eulogio Valentín Gómez , Gabriela Correa Carvalho, Tais Maria Bauab, Marlus Chorilli and Javier Pemán.

In view of the increase in infectious diseases, microbial resistance, and the emergence of new lethal species, this manuscript presents important results of pharmaceutical nanotechnology in the development of nanoemulsions loaded with amphotericin B for the treatment of Candida auris, a new resistant species.

The manuscript has been completely rewritten to make this easier to understand. For this reason, we omitted the change control because the totality of the article is different. We hope that this revision improves the quality of the manuscript for publication. This manuscript presents promising and encouraging results since the nanoemulsions loaded with amphotericin B showed better antifungal activity against planktonic cells, biofilms, and infections in an alternative in vivo model of Galleria mellonella.

Yours sincerely,

Dra. Alba Ruiz-Gaitán MD, PhD

On behalf of all the authors of the article referred to above

Review 4.

Comments and Suggestions for Authors

  • This study aimed to analyze the in vitro and in vivo antifungal activity of a nanoemulsion with amphotericin B (NE) against planktonic cells and biofilm of clinical isolates of C. auris belonging to four different clades.  Overall, the article contains experimental findings that can be interesting for readers of Microorganisms. The authors should address some issues to improve the manuscript.

Dear reviewer, thank you for reviewing the manuscript. the manuscript has been completely rewritten to make it more easily readable.

2) Page 2, lines 50-51: Please, check 106 cells/hour. I believe the correct is 106 cells/h. Hour is time unit and represented by h. Please change hour to h here and throughout the manuscript.

Dear reviewer, We have corrected this information.

3) Page 3, lines 104-105: Was the supernatant removed and treated? 

Dear reviewer, non-adherent cells (supernatant) were washed and removed and adhered cells in microplates were treated.

We have corrected the methodology in the manuscript, Section 2.3.1.

4) Page 3, line 196: AmP is not defined. Please, check it.

Dear reviewer, we have corrected this information. AmP = Ampicillin

5) The methodology for histology of the larvae was not provided. Authors should also add methodology for granuloma quantification.

I did not understand the statistical analysis reported by authors. Anova two way should be used to make comparisons within-groups over time and among groups at each experimental period. 

Dear reviewer, we have corrected this information. The objective was to evaluate the difference between groups of untreated larvae (PBS+AmP or NE) with the treated group (NEA and AmB) over time and to evaluate the difference between groups on each day of treatment.

6) Page 4, lines 157-158: In methodology, authors report that MIC was determined by inactivation of 90% of yeasts. However, they provide MIC50. Regarding MIC50, all strains show lower MIC for NE than AMB. Please, verify.

Statistical analyses were performed using GraphPad PRISMA 8.0. For the in vitro (biofilm) assays, one-way analysis of variance (ANOVA) was used to compare treatment with growth control. For the in vivo assays, two-way ANOVA was used to determine intragroup differences and to compare between groups over five days of treatment. Tukey's test correction was used for multiple comparisons between the groups. A p-value <0.01 is indicated in each statistical analysis.

Results

3.1 Determination of the Minimum Inhibitory Concentration (MIC)

AmB and NEA MICs are shown in Table 1. The MIC90 of NEA was considerably lower than that of AmB for VEN C6 (MIC90 1.25 vs 0.31 µg/mL), JAP 1 (MIC90 0.31 vs 0.038 µg/mL) and SP96 (MIC90 0.62 vs 0.038 µg/mL). Additionally, NEA MIC50 values were lower. No fungal growth was observed in any of the control groups.

7) Page 5, fig 1. Please, check y-axis of fig. 1B. Viability should be replaced by metabolic activity.

Figure corrected and added to manuscript.

8) Pages 5-6, lines 190-200: Here, authors evaluated biofilm formation. Differences among strains were attributted to the presence of EPS but I am not sure if EPS could explain these data. In methodology, authors described that non-adhered cells were treated. Is there any reference to support the lower EPS presence for JAP 1 and SP96 in biofilm formation? Indeed, the reference 22 should be used to discuss mature biofilms. 

Dear reviewer, new information has been added to justify the better behaviour of the free drug for strains JAP 1 and SP96.

“During biofilm formation, yeast produces EPS in the phases of surface attachment, cell recognition, and proliferation. In the beginning of formation, the percentage of EPS is low, making the yeast susceptible to the external environment, the host immune system and the activity of antifungal drugs.  [15,25,26] . In our study, we obtain high AmB activity against C. auris biofilm from Japan (JAP 1) and Spain (SP96) strains, which could be related to the lower presence of EPS, improving the inhibition of yeast growth by the drug. It is known that the East Asian strains, such as the Japanese strain (JAP 1) used in this study, have some peculiarities, such as increased susceptibility to fluconazole and low biofilm production. Due to these characteristics, these strains rarely cause systemic infections or large-scale outbreaks [24]. In our study, NEA inhibits biofilm formation by strains from India (InP13) and Venezuela (VEN C6), especially at lower concentrations. Similarly, a previous study evaluating the activity of NEA against C. auris strain CDC B11903 showed that NEA exhibited better inhibition of biofilm formation compared to AmB at lower concentrations (0.19 to 0.02 µg/mL) [15]. However, this study had the limitation of containing only one clinical isolate from each clade. To confirm these results, a large-scale study is needed”.

9) Pages 8, fig.3: Authors should normalize the data. For InP13 (fig. 3A) this is ok, but for other strains there are some differences in the initial infection that may lead to a misinterpretation of the data. In addition, I was not able to understand the letters and b. The legend is not complete.

Dear reviewer, during the antifungal assay, we selected three larvae from each group after infection to determine the percentage of yeast after infection. These results were included in the manuscript and considered as zero time in the images. From these results all groups started with a similar fungal load. Furthermore, we add this information in the methodology.

10) Page 9, fig. 4: Once again, I did not understand the letters a and b. What do the statistically significant differences refer to?

The presence of letters indicates a statistical difference comparing the infection control group (PBS+AmP or NE) with the treated group (AmB and NEA). The letter "b" indicates that NEA is also different from AmB (with NEA and AmB different from the control group).

11) Page 10, line 299: Were immune cells labeled "h" as hematocytes?

 “h” was modificated to hemocytes

12) Figs 5-12: Please, add "original" before magnification. The micrographs should contain a bar since the magnification changes according to the image size. PAS was not defined (figs. 9-12).

At the suggestion of three reviewers, we have removed many images. I hope that this new way of organising the images will improve their visualisation. We have added scale to the images (Figures 5-8).

13) Fig 13: The data should contain standard deviation. Please, add the number of samples in the legend. Indeed, the number of samples should be added in all figure legends of quantitative data.

Dear reviewer, we have added the SD and number of samples.

14) Please, revise the text carefully. There are some typos (e.g., ESP instead EPS).

Dear reviewer, we have made the corrections.

Round 2

Reviewer 1 Report

 Authors responded to most of my concerns. I have only few doubts. By "origin of species" I would like ask if they are taken from patient or they are standardized species. Table 1 - Please add information about the SD, and statistical analysis.

Author Response

Dear reviewer.

Thank you for reviewing the manuscript.

C. auris strains from other countries used were C. auris VPCI479/P13 (India, CLADE I - InP13), C. auris CBS10913 (Japan, CLADE II - JAP 1), C. auris CBS 15603 (Spain, CLADE III - SP96) and C. auris VEN C6 (Venezuela, CLADE IV), provided by the Dutch reference centre Centraalbureau voor Schimmelcultures (CBS).

2.  Information about the SD and statical analysis were incorporated into the manuscript.

Reviewer 4 Report

The authors have revised and improved the manuscript properly.

Author Response

Thank you for reviewing the manuscript. We are grateful for all your comments.